# A NIR-Activated and Mild-Temperature-Sensitive Nanoplatform with an HSP90 Inhibitor for Combinatory Chemotherapy and Mild Photothermal Therapy in Cancel Cells

**DOI:** 10.3390/pharmaceutics15092252

**Published:** 2023-08-31

**Authors:** Yingying Peng, Hanlin Jiang, Bifei Li, Yue Liu, Bing Guo, Wei Gan

**Affiliations:** 1Shenzhen Key Laboratory of Flexible Printed Electronics Technology, School of Science, Harbin Institute of Technology (Shenzhen), University Town, Shenzhen 518055, China; 19b958028@stu.hit.edu.cn (Y.P.); jianghljianghl@163.com (H.J.); 19b958029@stu.hit.edu.cn (B.L.); 21s058061@stu.hit.edu.cn (Y.L.); 2School of Chemistry and Chemical Engineering, Harbin Institute of Technology, Harbin 150001, China; 3Shenzhen Key Laboratory of Advanced Functional Carbon Materials Research and Comprehensive Application, Harbin Institute of Technology (Shenzhen), Shenzhen 518055, China

**Keywords:** BODIPY, mild photothermal therapy, NIR-II fluorescence imaging, combinatory therapy

## Abstract

Mild photothermal therapy (PTT) shows great potential to treat cancers while avoiding unwanted damage to surrounding normal cells. However, the efficacy of mild PTT is normally moderate because of the low hyperthermia temperature and limited light penetration depth. Chemotherapy has unlimited penetration but often suffers from unsatisfactory efficacy in view of the occurrence of drug resistance, suboptimal drug delivery and release profile. As a result, the combinatory of chemotherapy and mild PTT would integrate their advantages and overcome the shortcomings. Herein, we synthesized an NIR-activatable and mild-temperature-sensitive nanoplatform (**BDPII-gel@TSL**) composed of temperature-sensitive liposomes (TSL), heat shock protein 90 (HSP90) inhibitor (geldanamycin) and photothermal agent (**BDPII**), for dual chemotherapy and mild PTT in cancer cells. **BDPII,** constructed with donor-acceptor moieties, acts as an excellent near-infrared (NIR) photothermal agent (PTA) with a high photothermal conversion efficiency (80.75%). **BDPII**-containing TSLs efficiently produce a mild hyperthermia effect (42 °C) under laser irradiation (808 nm, 0.5 W cm^−2^). Importantly, the phase transformation of TSL leads to burst release of geldanamycin from **BDPII-gel@TSL,** and this contributes to down-regulation of the overexpression of HSP90, ensuring efficient inhibition of cancer cell growth. This research provides a dual-sensitive synergistic therapeutic strategy for cancer cell treatment.

## 1. Introduction

Nowadays, treatment of malignant tumors is still a big challenge. This is a major cause of shorter life spans for patients around the world [1,2,3]. So far, chemotherapy is an effective therapeutic strategy to destroy tumor cells [4,5,6]. However, most reported chemotherapeutic agents are incapable of selectively anchoring at the tumor site; thus, they often induce unwanted destruction to normal tissue [7]. Moreover, tumor cells rapidly generate drug resistance after chemotherapy, which also leads to a limitation for this promising therapy [8,9]. Recently, a nano-sized tumor-anchored dual therapy system based on chemotherapeutics has demonstrated an advantage in the precise delivery of drugs to the tumor site and induced satisfactory tumor ablation with negligible drug resistance [7,10]. In particular, the positively triggered drug release system can overcome the shortage of inefficient drug release in a passive drug release system and is thus highly desired [11]. 

Recent studies have shown that light-controlled therapy with spatiotemporal controllability is a precision strategy with minimized side effects, and the light energy could also be a triggering source to induce drug release in light-stimulus drug delivery systems [12,13,14]. For instance, the active and controllable generation of ROS (reactive oxygen species) in PDT (photodynamic therapy) and hyperthermia in PTT (photothermal therapy) can be triggered by light irradiation and is guaranteed to be highly efficient at the spatiotemporal destruction of tumor cells [13,13,14,15,16,17,18,19,20,21,22,23]. The light irradiation for PTT can take advantage of NIR (near-infrared window, λ = 700–1100 nm) light, which can achieve much higher penetration than visible light sources [13,24,25,26,27,28]. Notably, most PTT is effective in surficial tumors and hardly shows efficacy in tumor cells far away from the light irradiation site owing to the intrinsically limited penetration depth for light in living tissues. Moreover, the NIR light in PTT also could be a stimulus source to positively trigger drug release in chemotherapy.

Normally, the NIR-triggered PTT strongly relies on hyperthermia (˃50 °C) to completely eradicate cancer cells, so that the resultant heating diffusion often brings in side-effects to surrounding normal tissues [20,23,29,30,31]. The emergence of mild PTT (~45 °C) enables a promising strategy to inhibit cancer cell growth without damage to healthy tissues through different activation of tumor and normal cells [32,33,34]. However, the low-temperature property in mild PTT may also produce insufficient and heterogeneous heat to the tumor, which causes subsequent survival and self-recurrence of tumor cells [34]. More importantly, the expression of the 90 KDa heat shock protein (HSP90) is sharply increased even with a temperature lower than 42 °C so that the therapeutic efficacy of mild PTT may be further limited [17,20,21,23,26,29,35,36,37,38]. Therefore, combinatory mild PTT and other therapeutics to overcome this drawback could be a better choice for optimal cancer treatment.

As one of the HSP families, HSP90 displays an indispensable role in cancer cell proliferation, apoptosis and differentiation [39,40,41,42]. Once the tumor cells are exposed to heat stimulation in PTT, HSP90 is overproduced in tumor cells to keep the cells alive. Therefore, it is also a promising targeting site to induce complete tumor cell ablation and boost the therapeutic efficacy in mild PTT [29,35,43]. In this contribution, we constructed an NIR-triggered dual-sensitized nanoplatform (**BDPII-gel@TSL**) based on a temperature-sensitive liposome (TSL), in which both HSP90 inhibitor (geldanamycin) and photothermal agent (aza-BODIPY) were encapsulated into TSL to realize a chemo–photothermal synergistic therapy to effectively inhibit tumor growth at mild temperature (42 °C) (as shown in Figure 1). The construction of TSL is widely used to improve the drug release in thermotherapy through external local heat trigged by light irradiation of photothermal agent (PTA) [44,45,46,47]. A recent study indicates that TSL with a composition of DPPC/DSPC/DSPE-PEG-2000 is stable in biological media at 37 °C while undergoing a gel-to-liquid phased transition and inducing a 60% drug release at mild temperature (42 °C) (Figure 1a). Thus it is a potential encapsulating shell applicable in mild PTT [11,47]. As an excellent PTA in this nano-system, aza-BODIPY displays strong NIR absorbance and near-infrared second window (NIR II) emission, which enables it to have a high photothermal conversion in PTT (as shown in Figure 1b) [48,49,50,51,52]. In addition, geldanamycin can specifically bind to the HSP90 through the amine group to inhibit the overexpression of the protein and improve the heat sensitivity of cancer cells [40,53,54,55]. Additionally, the anticancer process of BDPII-gel@TSL NPs was turned on through the irradiation of 808 nm with a low power density (0.5 W cm^−2^), where the photothermal effect of BDPII in NPs was excited to generate a mild temperature (42 °C) and induce a phase transition of TSL to positive release the geldanamycin and BDPII, activating the chemo–photothermal dual synergistic therapy. Then, the heat sensitivity of cancer cells was enhanced by the inhibition of HSP90, which enables considerable tumor cell death at a mild temperature produced by BDPII.

## 2. Materials and Methods

### 2.1. Materials

The chemicals including 1-(4-hydroxyphenyl)ethan-1-one, 4-(diethylamino)benzalde hyde, Potassium hydro-xide (KOH), methanol (MeOH), Nitromethane (MeNO_2_), 1,2-diethylamine (DCE), ammonium acetate (NH_4_OAC), n-butanol (n-BuOH), boron (tri) fluoride etherate (BF_3_Et_2_O), *N*, *N*-Diisopropylethylamine (DIPEA), dichloromethane (DCM), petroleum ether (PE), ethyl acetate (EA), cyclohexane, 1,2-dipalmitoyl-*sn*-glycero-3-phosphocholine (DPPC), 1,2-distearoyl-*sn*-glycero-3-phosphocholine (DSPC), 1,2-distearoyl-*sn*-glycero-3-phospho ethanolamine-N-PEG2000 (DSPE-PEG_2000_), methylthiazolyldiphenyl-tetrazolium bro mide (MTT), Calcein-AM/PI assay kit, 4-[(E)-2-[(3Z)-2-chloro-3-[2-(2,6-diphenyl thiopyran-4-ylidene)ethylidene] cyclohexen-1-yl]ethenyl]-2,6 diphenylthiopyrylium, tetrafluorobo-rate (IR1061) were purchased from Sigma-Aldrich and used without further purification. Dulbecco’s modified essential medium (DMEM), Fetal bovine serum (FBS), Streptomycin-penicillin and phosphate buffer saline (PBS) were purchased from Gibco (Life Technologies, Carlsbad, CA, USA). Deionized water (18.2 MΩ·cm) was purified with a water purification system (WP-UP-UV-20) from Sichuan Water Technology Development Co. Ltd. (Chengdu, China). All other chemicals were used as received.

### 2.2. Synthesis and Characterization

The NMR and ESI-MS spectra for BDPII are measured by a Bruker AV-400 spectrometer and LTQ Orbit rap XL instruments, respectively. The synthetic routes and characterization results of BDPII are displayed in Figure 1a and Appendix A.

*Synthesis of (E)-3-(4-(diethylamino)phenyl)-1-(4-hydroxyphenyl)prop-2-en-1-one* (**Compound 3**). 1-(4-hydroxyphenyl)ethan-1-one (**compound 1**, 5 g, 36.75 mM), 1-(4-(diethylamino)phenyl)ethan-1-one (**compound 2**, 6.15 g, 36.75 mM) and KOH (6.18 g, 110.25 mM) were dissolved in MeOH and stirred at room temperature for 4 h, followed with evaporation under reduced pressure and purification with silica gel column chromatography using PE/ DCM (*v/v*, 8:1) as the eluent to provide the bright red products with the yield of 86%. ^1^H NMR (400 MHz, DMSO) *δ* 8.01 (d, *J* = 8.7 Hz, 2H), 7.64 (d, *J* = 8.8 Hz, 2H), 7.58 (d, *J* = 3.6 Hz, 2H), 6.87 (d, *J* = 8.7 Hz, 2H), 6.69 (d, *J* = 8.9 Hz, 2H), 3.41 (q, *J* = 7.0 Hz, 4H), 1.11 (t, *J* = 7.0 Hz, 6H). ^13^C NMR (101 MHz, DMSO) *δ* 186.89 (s), 161.61 (d, *J* = 18.7 Hz), 149.32 (s), 144.04 (s), 130.91 (d, *J* = 18.1 Hz), 129.99 (s), 121.50 (s), 115.62 (s), 115.27 (d, *J* = 9.0 Hz), 111.22 (s), 43.90 (s), 12.59 (s). ESI (m/z): calculated for C_19_H_21_NO_2_^+^, [M+H]^+^ 296.1956, found 296.1953.

*Synthesis of 3-(4-(diethylamino)phenyl)-1-(4-hydroxyphenyl)-4-nitrobutan-1-one* (**Compound 4**). The compound 3 (2 g, 6.78 mM), MeNO_2_ (2.885 g, 33.90 mM) and DEA (3.76 g, 51.40 mM) were dissolved in EtOH and stirred at 40 °C. After 24 h, the EtOH were removed by evaporation, the crudes were washed by brine and extracted with DCM. The organic fraction was dried over Na_2_SO_4_, followed with evaporation under reduced pressure and purification with silica gel column chromatography using PE/ EA (*v*/*v*, 8:1) as the eluent to provide the red oil products with yield of 65%. ^1^H NMR (400 MHz, CDCl_3_) *δ* 7.88–7.79 (m, 2H), 7.13-7.03 (m, 2H), 6.88–6.80 (m, 2H), 6.63–6.54 (m, 2H), 4.76 (ddd, *J* = 12.1, 6.7, 1.7 Hz, 1H), 4.61 (ddd, *J* = 12.1, 7.9, 1.7 Hz, 1H), 4.20–4.01 (m, 2H), 3.33–3.27 (m, 5H), 1.16–1.08 (m, 7H). ^13^C NMR (101 MHz, CDCl_3_) *δ* 196.31 (s), 161.19 (s), 147.32 (s), 130.74 (s), 129.29 (s), 128.30 (s), 125.40 (s), 115.56 (s), 112.09 (s), 80.18 (s), 77.40 (s), 77.40–76.85 (m), 76.76 (s), 44.32 (s), 41.52 (s), 38.80 (s), 12.56 (s). ESI (m/z): calculated for C_20_H_24_N_2_O_4_^+^, [M+H]^+^ 357.2117, found 357.2121.

Synthesis of (Z)-4-(3-(4-(diethylamino)phenyl)-2-((3-(4-(diethylamino)phenyl)-5-(4-hydroxyphenyl)-1H-pyrrol-2-yl)imino)-2H-pyrrol-5-yl)phenol (**Compound 5**). Compound 4 (0.5 g, 1.40 mM) and NH_4_OAC (1.62 g, 21.05 mM) were dissolved into n-BuOH and reacted under reflux at 110 ℃ for 24 h. After cooling to the room temperature, the crude products were filtered, washed with EtOH and dried by evaporation under reduced pressure, then, followed by purification with silica gel column chromatography using PE/EA (*v/v*, 2:1) as eluent to provide the dark bule products with yield of 75%. ^1^H NMR (400 MHz, DMSO) δ 8.00 (d, *J* = 8.8 Hz, 4H), 7.86 (d, *J* = 8.6 Hz, 4H), 7.23 (s, 2H), 6.98 (d, *J* = 8.6 Hz, 4H), 6.73 (d, *J* = 8.9 Hz, 4H), 3.43 (dd, *J* = 13.8, 6.8 Hz, 8H), 1.15 (t, *J* = 7.0 Hz, 12H). ^13^C NMR (101 MHz, DMSO) δ 160.01 (s), 154.12 (s), 150.69 (s), 148.76 (s), 147.61 (s), 130.51 (s), 128.56 (s), 123.49 (s), 116.78 (s), 111.43 (s), 111.19 (s), 44.33 (s), 13.05 (s). ESI (m/z): calculated for C_40_H_41_N_5_O_2_^+^, [M+H]^+^ 624.3690, found 624.3694.

Synthesis of 4,4′-(1,9-bis(4-(diethylamino)phenyl)-5,5-difluoro-5H-4l4,5l4-dipyrrolo [1,2-c:2′,1′-f][1-3,5]triaza borinine-3,7-diyl)diphenol (**BDPII**). Compound 5 (0.5 g, 0.80 mM) and DIPEA (1.036 g, 8 mM) were dissolved in dry DCM and stirred for 30 min at 0 °C. Then, the BF_3_OEt_2_ was added into the mixture and reacted with room temperature for 24 h. The crudes were washed with water three times and dried over Na_2_SO_4_, followed by the purification with silica gel column chromatography using PE/ EA (*v/v*, 2:1) as eluent to provide the dark blue products with yield of 45%. ^1^H NMR (400 MHz, CDCl_3_) δ 8.05 (d, *J* = 8.7 Hz, 4H), 7.93 (d, *J* = 8.4 Hz, 4H), 6.84 (d, *J* = 8.4 Hz, 4H), 6.77–6.67 (m, 6H), 3.43 (dd, *J* = 13.6, 6.7 Hz, 8H), 1.22 (d, *J* = 7.1 Hz, 13H). ^13^C NMR (101 MHz, CDCl_3_) δ 157.97 (s), 156.76 (s), 148.37 (s), 145.08 (s), 142.92 (s), 131.20 (s), 129.96 (s), 124.82 (s), 120.57 (s), 115.66 (s), 114.36 (s), 111.36 (s), 29.75 (s), 12.76 (s). ESI (m/z): calculated for C_40_H_40_BF_2_N_5_O_2_^+^, [M]^+^ 671.3662, found 671.3664.

### 2.3. UV-Vis and Fluorescence Spectra Characterization

The UV-Vis and Fluorescence spectra were monitored with 1 cm quartz cuvette through Shimadzu (Suzhou, China) UV-1800 spectrophotometer and F-7000 spectrophotometer (HATACHI, Shizuoka, Japan), respectively.

### 2.4. Molar Extinction Coefficient Measurement

Initially, the absorbance of BDPII solution with different concentration were recorded using the UV-1800 spectrophotometer. the molar extinction coefficient (*ε*) for BDPII was calculated by the Beer-Lambert Law: A=εlcwhere *A* indicates the absorbance of BDPII at 808 nm, *l* and *c* represent the optical path (1 cm) and the concentration of BDPII solution, respectively. 

### 2.5. Preparation of TSL Nanoparticles

The nanoparticles of BDPII@TSL and BDPII-gel@TSL were prepared with ultrasonic method [56]. Briefly, DPPC, DSPC and DSPE-PEG2000 (molar ratio, 80:15:5) were dissolved in chloroform and dried under vacuum to remove the chloroform [11]. Then, 10 mL BDPII, geldanamycin and BDPII/geldanamycin aqueous solution were added to form lipid film, flowed by a sonication for 40 min at 60 °C. After cooling to room temperature, these solutions were filtrated through syringe membrane with a pore size of 0.8 µm to remove the undesirable aggregates and stored at 4 °C for further measurement. 

### 2.6. Quantum Yield Test

The quantum yield (*Φ*) of BDPII, BDPII@TSL and BDPII-gel@TSL NPs was determined by the previous reported method [57]. A typical NIR-II dye IR1061 was used as a standard with the NIR-II quantum yield of 0.75%. To obtain the value of QY, the IR-1061, BDPII, BDPII@TSL and BDPII-gel@TSL NPs were diluted through water with the 808 nm absorbance of 0.02, 0.04, 0.06, 0.08 and 0.10, after that, the emission spectra for these samples were detected to achieve the emission integration of peak area at the range of 900–1300 nm. Then, a slope of linear relation between the emission integration of peak area and the absorbance at 808 nm were determined to calculate the value of QY. The value of *Φ* for BDPII, BDPII@TSL and BDPII-gel@TSL were achieved by the following equation: $$\Phi_{sample}=\Phi_{ref}*\frac{{Slope}_{sample}}{{Slope}_{ref}}*{\left(\frac{n_{sample}}{n_{ref}}\right)}^{2}$$
where the values of *Slope_sample_* and *Slope_ref_* are determined by the linear relationship between the integration of emission peak area and the absorbance at 808 nm. *n_sample_* and *n_ref_* indicate the refractive index of the solvent in the sample and reference, respectively. Here they are both the refractive index of water.

### 2.7. Photothermal Conversion Efficiency Calculation

The photothermal conversion efficiency (PCE, *η*) for BDPII@TSL and BDPII-gel@TSL NPs in water were measured according to the reported works [28,58]. First, these NPs were irradiated by 808 nm laser with power of 1 W cm^−2^ for 15 min. Then, these samples were cooling down to room temperature with the temperature recorded by an infrared camera. The value of *η* for BDPII@TSL and BDPII-gel@TSL were calculated through the following equation: (1)$$\eta =\frac{hA\left(T_{Max}-T_{Surr}\right)-Q_{Dis}}{I\left(1-10^{-A}\right)}$$
where *h* and *A* indicate the heat transfer coefficient and surface area of the container, respectively. *T_Max_* and *T_Surr_
*represent the maximum steady-state (62.73 °C for BDPII-gel@TSL and 62.51 °C for BDPII@TSL) and room temperature, *I* represents the laser power (1 W cm^−2^), *A* is absorbance of the specimen at 808 nm, *Q_Dis_* is the heat dissipation from the laser mediated by the solvent and the container, which could be calculated by Equation (2), the value of *hA* could be calculated through the Equation (3).
(2)$$Q_{Dis}=\frac{mc(T_{Max\left(water\right)}-T_{Surr})}{\tau_{s(water)}}\;$$
where *T_Max(water)_* is 28.9 °C, the value of *τ_s_* for water is 297 s upon 808 nm irradiation.
(3)$$\tau_{s}=\frac{mc}{hA}\;\;\;\;\;\;\;\;$$
where *m* is mass of the aqueous solution (1.0 g) and *c* is the heat capacity of water (4.2 J g^−1^), *τ_s_* is the time constant which could be calculated by Equations (4) and (5):(4)$$t=-\tau_{s}\ln\left(\frac{T_{RT}-T_{Surr}}{T_{Max}-T_{Surr}}\right)$$
(5)$$\theta =\frac{T_{RT}-T_{Surr}}{T_{Max}-T_{Surr}}$$*T_RT_* is the real temperature in this cooling process, *θ* is the driving force temperature. 

### 2.8. Cell Incubation Experiment

HeLa (America Type Culture Collection) cells were incubated in a DMEM containing 10% FBS and 1% antibiotics composed by penicillin (100 units/mL)-streptomycin (100 μg/mL) using an incubator with a humidified 5% CO_2_ at 37 °C.

### 2.9. In vitro Confocal Laser Scanning Microscopy (CLSM)

In the live/dead cell co-staining assay, HeLa cells were seeded in 35 mm confocal chamber with a density of 7 × 10^4^ for 24 h. The culture medium was placed with fresh medium containing 40 μM NPs to incubate for another 3 h. Then, the cells were irradiated under white light (0.5 W/cm^2^) for 15 min, followed by the addition of Calcein-AM (3 μM) and PI (4 μM) for 30 min, the cells were washed with PBS and immediately imaged using CLSM. 

### 2.10. In Vitro Dark & Light Cytotoxicity

The cell viability was estimated by MTT assay. The HeLa cells were seeded in 96-well plate with a density of 7 × 10^4^ and incubated for 24 h. Then, the medium was replaced by fresh medium containing different concentration of NPs and the cells were cultured for another 24 h at 37 °C. The culture medium was then discarded, followed by an addition of MTT solution with the concentration of 0.5 mg/mL. After 4 h incubation, the cells were irradiated with white light (0.5 W/cm^2^, 20 min) and further incubated for 24 h at 37 °C. Finally, the MTT solution was gently removed, 100 μM DMSO was added to dissolve the crystals. Then, the absorbance for each plate at 540 nm was measured through a Bio-Rad microplate reader. The cell viability was calculated by the following equation [59]:$$\mathrm{Cell}\;\mathrm{viability}\left(\%\right)=\left(\;\frac{{OD}_{treated}-{OD}_{blank}}{{OD}_{control}-{OD}_{blank}}\right)\times 100\;\%$$

For the dark cytotoxicity sample, the irradiating step was absent and rest part of the experiments was unchanged.

## 3. Results and Discussion

### 3.1. Synthesis and Characterization of BDPII

PTAs with absorbance in NIR window are widely used in PTT for their high penetration ability and excellent optical performance, preferably at 808 nm for their high light harvesting and deep tissue penetration [24,25,26,27,28,60]. Unfortunately, NIR-triggered PTT is usually under the restriction of poor photostability, depressed light absorption capacity and low photothermal conversion efficiency (PCE) of PTAs [26,28,58,61]. Among numerous reported NIR photothermal agents, aza-boron-dipyrrome-thenes (aza-BODIPY) derivatives are photostable materials with a high molar absorption coefficient at the NIR window, a favorable non-radiative excitation decay pathway to efficiently produce photothermal energy, and adjustable photophysical properties and easy functionalization, all of which endow great potential for aza-BODIPY-based PTAs in PTT application [28,50,62,63]. The donor-acceptor (D-A) molecular engineering favors strong intramolecular charge transfer (ICT) and a narrow energy band gap [64], which facilitate a redshift of absorption to the NIR window and efficient fluorescent quenching to improve the PTT performance [65,66].

In this work, we used the electron-rich *N*,*N*-diethylaniline groups as electron donors and the classical aza-BODIPY electron-withdrawing cores as electron acceptors to construct the NIR photothermal agent (**BDPII**). The synthetic routes and characterization (NMR and ESI-Mass spectrum) of **BDPII** were shown in Figure 1a and Appendix A, respectively. Briefly, **BDPII** was enlarged through the conjugation of *N*,*N*-diethylaniline and phenol groups on 3,5-position and 1,7-position of the aza-BODIPY core, respectively. The large conjugating system of **BDPII** enabled its high extinction coefficient (*ε*), and the strong charge transfer along the D-A backbone endowed a redshift to the NIR window as well as desirable PCE in the PTT application. As shown in Figure 1b and Table 1, the **BDPII** has two absorbance peaks visible in the NIR region, in which the short absorbance peak at about 639 nm is ascribed to the π-π* transition, and the long wavelength at about 808 nm is due to intramolecular charge transfer (ICT) effect of BDPII [64,67,68]. Additionally, the BDPII also exhibits an excellent extinction coefficient at 808 nm of 30,980 mol^−1^ cm^−1^ (Figure 1c,d). Benefiting from the strong ICT effect of **BDPII**, the emission spectrum is enlarged to the NIR II window in the range of 900–1000 nm (Figure 1e). However, it is impressive that it also demonstrates strong fluorescence quenching in water with a fluorescence quantum yield (*Φ*) of 0.56% (Figure 1f,g and Table 1) [69,70,71] in the measurements using a typical NIR II material IR 1061 (75%) as a standard (Appendix A). As indicated in Figure 1h, with the increase in solvent polarity, the fluorescent intensity of **BDPII** shows a notable decrease, and the emission wavelength presents a redshift, all of which demonstrate a strong solvatochromic effect for **BDPII** and result in its weak emission as shown in Figure 1e [72,73,74].

### 3.2. Preparation of **BDPII-gel@TSL** NPs

Generally, a PTA with good water dispensability is required for biological application. Therefore, we used an ultrasonic method to formulate water-dispersible **BDPII-gel@TSL** with raw materials, including **BDPII**, geadanamycin and TSL, in which the TSL were prepared with the constituents of DPPC/DSPC/DSPE-PEG_2000_ by a molar ratio of 80:15:5 [11,56]. BDPII-gel@TSL was filtered through a syringe membrane with a pore size of 0.8 µm to remove the undesirable aggregates to achieve the relatively uniform-sized product. Owing to the hydrophobic core of **BDPII** and geldanamycin, as well as the hydrophilic shell of TSL, the **BDPII** and geldanamycin are inclined to locate in the phospholipid bilayer region (Figure 2a), in which the contents in TSL NPs were easily released upon heat triggering. Transmission electron microscopy (TEM) and dynamic light scattering (DLS) results showed spherical morphologies for **BDPII-gel@TSL** NPs with an average diameter of about 137 nm (Figure 2b and Appendix A). According to the time-dependent diameter monitored by DLS shown in Figure 2c and Appendix A, they remained stable in water, PBS and DMEM over two weeks of storage at 4 °C, indicating excellent stability for potential biological application. In order to investigate the synergistic effect between **BDPII** and geldanamycin in mild PTT conditions, **BDPII** was solely encapsulated in TSL to form **BDPII@TSL**, according to the same preparing protocol as for **BDPII-gel@TSL**. The results from DLS and TEM demonstrate spherical morphologies for BDPII@TSL NPs with an average diameter of approximately 116 nm in water (Appendix A). Furthermore, it also shows good stability in water, PBS and DMEM with 4 °C storage for 15 days (Appendix A). 

The fluorescence emission of **BDPII-gel@TSL** NPs was recorded and displayed in Figure 2d. Once **BDPII** molecules were encapsulated into TSL, the fluorescence intensity was notably enhanced, and the value of ***Φ*** for **BDPII-gel@TSL** (6.85%) was ten times higher than that for **BDPII** (0.56%) (Figure 2e,f and Table 1), which is attributed to the solvatochromic effect of **BDPII** as shown in Figure 1h. That is to say, with the BDPII molecules successfully loaded in TSL to form **BDPII-gel@TSL** NPs, the hydrophobic phospholipid bilayer region of TSL provides a non-polar environment for BDPII to exhibit strong emission. Therefore, the heat-triggered release of **BDPII-gel@TSL** NPs can be determined through fluorometry at 42 °C. As shown in Figure 2g, the fluorescence intensity of **BDPII-gel@TSL** NPs gradually decreases upon the irradiation for 0 min to 5 min (0.5 W cm^−^^2^, 808 nm), indicating the effective release of **BDPII** from the non-polar phospholipid bilayer of TSL into the polar medium (aqueous solution).

### 3.3. Photothermal Performance of **BDPII-gel@TSL** NPs 

Encouraged by the excellent NIR absorption ability of **BDPII**, the photothermal performance for **BDPII-gel@TSL** NPs was further investigated upon 808 nm irradiation and shown in Figure 3. The temperature of **BDPII-gel@TSL** NPs in PBS solution displayed a dramatic increase upon 1 W cm^−2^ laser irradiation. Within 15 min of irradiation, the temperature of **BDPII-gel@TSL** NPs reached a temperature as high as 62.73 °C (Figure 3a). The corresponding infrared photos were recorded, as shown in Figure 3b. It was found that the temperature increase has a positive dependence on the concentration and laser power density (Figure 3c,d). Notably, the temperature could increase to 43 °C after 808 nm laser irradiation with 0.5 W cm^−2^ for 15 min (Figure 3d), which endows **BDPII-gel@TSL** NPs with potential in mild PTT application. The **BDPII-gel@TSL** NPs demonstrated no obvious photobleaching effect after multiple heating–cooling cycles with laser irradiation (Figure 3e), suggesting excellent photothermal stability. According to the temperature curve shown in Figure 3a, the photothermal conversion efficiency (PCE) of **BDPII-gel@TSL** NPs was calculated to be 80.75% (Figure 3f), which is quite high among previously reported PTAs [28,58,75], indicating its potential to be applied in mild PTT. The **BDPII@TSL** NPs exhibited an excellent PCE with a value of 78.70% (Appendix A), suggesting that the loading of geldanamycin hardly affects the optical properties of **BDPII**.

### 3.4. In Vitro Mild PTT Application

Given the good photothermal conversion capability and high photostability of the **BDPII-gel@TSL** NPs, the tests, including cytotoxicity and combinatory photothermal and chemotherapeutic efficacy, were examined via standard MTT assay. The experimental groups included the geldanamycin group, **BDPII@TSL** group and **BDPII-gel@TSL** group. Initially, upon NIR irradiation (808 nm) with a lower power density (0.5 W cm^−2^) for 30 min, HeLa cells in both the geldanamycin and **BDPII@TSL** groups displayed moderate cytotoxicity at mild temperature (42 °C), as shown in Figure 4a. It was calculated that the values of IC50 for the geldanamycin and **BDPII@TSL** groups were as high as 53.27 μM and 43.16 μM (Figure 4b), respectively. As compared to the geldanamycin and **BDPII@TSL** groups, the **BDPII-gel@TSL** group displayed much lower cell viability under the same conditions, with a low IC50 value of 24.36 μM (Figure 4b). Reasonably, after exposure to light irradiation, the moderate temperature (42 °C) produced by the **BDPII** would initially destroy the nanostructure of **BDPII-gel@TSL** and cause the burst release of **BDPII** and geldanamycin. This synergistic effect of photothermal therapy and chemotherapy with a burst release manner led to the high ablation efficacy on tumor cells (Figure 4a). Under dark conditions, as shown in Figure 4c, the cell viability for **BDPII@TSL** displayed negligible change and maintained a more than 90% survival rate, indicating good biocompatibility for **BDPII@TSL**. It was found that the cell viability for geldanamycin showed an apparent decrease with an increase in concentration, which is ascribed to the intrinsic chemotherapy of geldanamycin. It is known that geldanamycin, an HSP90 inhibitor, can cause proteasome-dependent degradation and simultaneously disrupt some critical signal ways to the growth of cancer cells through the bond between geldanamycin and N-terminal ATPase structural domain of HSP90 [76]. However, while the geldanamycin molecules were loaded into TSL to form **BDPII-gel@TSL** NPs, the bond of geldanamycin to HSP90 was restrained, in which the anticancer effect of geldanamycin in **BDPII-gel@TSL** NPs was significantly suppressed, and the decrease in cell viability of **BDPII-gel@TSL** was much minor. = In addition, the small loss of HeLa cell viability in Figure 4c was attributed to the slight leakage of geldanamycin from **BDPII-gel@TSL** NPs into the cells during endocytosis. 

In order to vividly examine the mild PTT efficacy in vitro, the calcein-AM and propidium iodide (PI) kit assay were used to visualize the HeLa cell mortality under NIR irradiation. Calcein-AM was used to label the living cells and can easily penetrate into the cell membrane and be hydrolyzed into the calcein structure to emit strong green fluorescence. Propidium iodide (PI) is a dead cell marker and can selectively be embedded into DNA dead cells and yield red fluorescence. As shown in Figure 4d, after exposure to NIR laser irradiation (808 nm, 0.5 W cm^−2^ for 30 min), the control group exhibited strong green emission with high cell activity, indicating that the low power density notably had no obvious influence on the cell viability. Owing to the sole therapeutic strategy of geldanamycin and **BDPII** as described in the MTT assay, the chemotherapy of geldanamycin and phototherapy of **BDPII@TSL** could merely compromise the cell viability less than 50%, and this means that the cells could be stained with both Calcein-AM and PI with green and red emission, respectively. In comparison, the cells incubated with **BDPII-gel@TSL** emit strong red fluorescence, indicating the synergistic photothermal and chemotherapeutic effects of the combination of **BDPII** and geldanamycin to effectively ablate cancer cells. Under dark conditions, the HeLa cells in both the control group and **BDPII@TSL** group exhibited strong green fluorescence, suggesting good biological compatibility of **BDPII@TSL**. The groups incubated with geldanamycin and **BDPII-gel@TSL** illustrate both green and red fluorescence without irradiation, indicating that the chemotherapy of geldanamycin partly inhibits tumor cell growth. Taken together, these results are consistent with the in vitro cytotoxicity results in Figure 4. Taken together, the NIR-activatable and mild-temperature-sensitive nanoplatform (BDPII-gel@TSL) is an efficient platform to ablate cancer cells in vitro.

## 4. Conclusions

In summary, we developed a TSL-coated dual-sensitized nanoplatform **BDPII-gel@TSL** loaded with PTA (**BDPII**) and HSP90 inhibitor (geldanamycin) for highly efficient combinatory chemotherapy and mild photothermal therapy of cancer cells. The synthesized **BDPII** exhibited functioned as an excellent PTA with NIR optical performance via donor-acceptor molecular engineering, while the PCE for **BDPII** could reach up to 80.73% upon the 808 nm irradiation. Importantly, **BDPII** nanoparticles could produce a mild temperature (42 °C) upon 0.5 W cm^−2^ of 808 nm laser irradiation to induce the phase transformation of TSL and stimulate the release of itself and geldanamycin from TSL. Furthermore, the triggered release of geldanamycin could initially down-regulate the overexpression of HSP90 and construct a photothermal-chemo synergistic therapy with **BDPII** to efficiently eradicate cancer cells under mild PTT conditions. This research points out new insights and directions for future improvement of mild PTT efficacy to effectively ablate tumor cells.

## Data Availability

Not applicable.

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
