# Peer review of "A NIR-Activated and Mild-Temperature-Sensitive Nanoplatform with an HSP90 Inhibitor for Combinatory Chemotherapy and Mild Photothermal Therapy in Cancel Cells"

_pharmaceutics, 2023, doi:10.3390/pharmaceutics15092252_

Round 1
Reviewer 1 Report
Though photothermal therapy is practiced in clinics it is a painful process in real time application where tumours are heated to until death at higher than body temperature. The nano particle formation via ultrasonication is very promising. Characterization is well supported with the DLS and TEM. The dual activity of this kind of compounds are very exciting. This manuscript could be accepted in it original form with small spelling mistakes. In conclusions. Spelling mistakes found in last sentence for example. chemo synergistic therapy with BDPII to efficiently eradicate cancel cells under mild PTT conditions.
Author Response
Author reply: We appreciate the reviewer`s positive evaluation to the manuscript and we revised the mistakes as suggested in the clear version of the manuscript.
Reviewer 2 Report
This manuscript by Peng et al describes results from studies of different nanoparticles (NPs) comprised of a photothermal agent (BDPII) encapsulated into a temperature-sensitive liposome (TSL), termed BDPII@TSL, vs. a combination of this formula with a chemotherapeutic, geldanamycin (BDPII-gel@TSL). The complete NPs promise to provide both photo- and chemotherapeutic effects with mild photothermal therapy. The authors thoroughly characterize the BDPII-gel@TSL NPs, including size distributions, diameter characteristics in different aqueous solutions, emission spectra and fluorescence emission. The NPs exhibit predicted photothermal heating, and little photobleaching. Most importantly, both the BDPII@TSL and BDPII-gel@TSCL exhibit photothermal efficacy in killing HeLa cells as a model of tumor cells, but the BDPII-gel@TSL also provides chemo-therapeutic cytotoxicity that, when combined with photothermal conversion, is improved compared to the chemotherapeutic effects of gelanamycin alone. The lack of this effect in dark conditions indicates that there should be little side-effect of the NPs, despite small amounts of drug released. Overall, the characterizations of the NPs are appropriate and support their predicted design, the data are well presented, and the positive in vitro effects warrant further studies of the BDPII-gel@TSL particles in vivo. However, there are some issues that need to be addressed.
1. There are numerous grammatical and spelling mistakes (or incorrect terms used) throughout the manuscript, so this reviewer highly recommends that a reader well-versed in English reviews the manuscript and fixes the numerous errors.
2. Scheme 1: The schematic for the generation of the BDPII-gel@TSL is very helpful, as is the diagram indicating actions of the nanoparticles upon irradiation and effects on tumor cells.
3. Under "Preparation of BDPII-gel@TSL NPs", 6th sentence, the authors discuss the diameter of the NPs in water, and indicate a photograph in Figure 2c and Table S1, but Figure 2c does not have a photo. Are they referring to the TEM photo shown in Fig. 2b? This is confusing.
Also, for the last sentence in this section, the text needs to be changed to indicate that the release of drug potentially could localize at high concentrations in cancer cells; it reads more as this was shown from analyses of cancer cells, but that has not been described.
4. Figure 2e, what are the dimensions for the different concentrations shown in the legend (0.02-0.1)? This should be noted either in the figure or in the legend text.
As noted in the comments, there are numerous grammatical mistakes (only a few words are misspelled) that should be fixed, so recommend a careful edit of the English and wording. However, the intent of most sentences is agreeable.
Author Response
Author reply: We appreciate the reviewer’s positive evaluation to the manuscript and the help on revising it.
- There are numerous grammatical and spelling mistakes (or incorrect terms used) throughout the manuscript, so this reviewer highly recommends that a reader well-versed in English reviews the manuscript and fixes the numerous errors.
Author reply: We revised the grammatical and spelling mistakes as suggested in the clear version of the manuscript.
- Scheme 1: The schematic for the generation of the BDPII-gel@TSL is very helpful, as is the diagram indicating actions of the nanoparticles upon irradiation and effects on tumor cells.
Author reply: Thanks. The scheme of generation of BDPII-gel@TSL was shown in Figure 2a. As shown in Figure 1h and 2d, the fluorescence of BDPII was highly enhanced once BDPII were encapsulated into TSL to form BDPII-gel@TSL owing to the solvatochronmic effect of BDPII, which was to say that the BDPII and geldanamycin were inclined to load into the hydrophobic phospholipid bilayer region which could provide a non-polar environment for BDPII to exhibit strong emission. And the anticancer mechanism of BDPII-gel@TSL upon 808 nm irradiation was shown Scheme 1b, in which the thermally induced release of geldanamycin could initially suppress the over expression of HSP90 and construct a chemo-photothermal synergistic therapeutic strategy to efficiently eradicate tumor cells in mild PTT.
- Under "Preparation of BDPII-gel@TSL NPs", 6th sentence, the authors discuss the diameter of the NPs in water, and indicate a photograph in Figure 2c and Table S1, but Figure 2c does not have a photo. Are they referring to the TEM photo shown in Fig. 2b? This is confusing. Also, for the last sentence in this section, the text needs to be changed to indicate that the release of drug potentially could localize at high concentrations in cancer cells; it reads more as this was shown from analyses of cancer cells, but that has not been described.
Author reply: Thanks for pointing this out, and we revised it in page 9 line 15 to 18 of the clear version of the manuscript.
(……According to the time-dependent diameter monitored by DLS shown in Figure 2c and Table S1, they remained stable in water, PBS and DMEM over two weeks storage at 4 ℃, indicating an excellent stability for potential biological application……)
- Figure 2e, what are the dimensions for the different concentrations shown in the legend (0.02-0.1)? This should be noted either in the figure or in the legend text.
Author reply: Thanks for the suggestion. The Figure 2e in this manuscript is used for the calculation of fluorescence quantum yield (QY) for BDPII-gel@TSL. To obtain the value of QY, the IR-1061 and BDPII-gel@TSL NPs were diluted through water with the 808 nm absorbance of 0.02, 0.04, 0.06, 0.08 and 0.10, after that, the emission spectra for these samples were detected to achieve the emission integration of peak area at the range of 900-1300 nm. Then, a slope of linear relation between the emission integration of peak area and the absorbance at 808 nm were determined to calculate the value of QY. And we added this experiment details in the Quantum yield test section of supporting information (page 6 line 15 to 21).
(……To obtain the value of QY, the IR-1061, BDPII, BDPII@TSL and BDPII-gel@TSL NPs were diluted through water with the 808 nm absorbance of 0.02, 0.04, 0.06, 0.08 and 0.10, after that, the emission spectra for these samples were detected to achieve the emission integration of peak area at the range of 900-1300 nm. Then, a slope of linear relation between the emission integration of peak area and the absorbance at 808 nm were determined to calculate the value of QY. The value of Φ for BDPII, BDPII@TSL and BDPII-gel@TSL were achieved by the following equation……)
Reviewer 3 Report
Peng et al, represent the Nanoplatform with HSP90 Inhibitor for Combinatory Chemo- and Mild Photothermal Therapy in Cancel Cells. The manuscript is well written with scientific sound. However, before publication following quires should be addressed.
-Introduction section should be more emphasis based on the used material and also give information related to BDPII-gel@TSL-cancer cell interaction to enhance anticancer functionalities.
-Provide the cell viability results of BDPII-gel@TSL against normal fibroblast or normal cells.
-Figure 4d, calculate fluorescence intensities and analyze for statistical difference.
Minor English editing is required.
Author Response
Introduction section should be more emphasis based on the used material and also give information related to BDPII-gel@TSL-cancer cell interaction to enhance anticancer functionalities.
Author reply: Thanks for pointing this out, and we added this emphasis as suggested in page 4 line 28 to page 5 line 1 to 6 of the clear version of the manuscript.
(……And the anticancer process of BDPII-gel@TSL NPs were turned on through the irradiation of 808 nm with a low power density (0.5 W cm-2), where the photothermal effect of BDPII in NPs was excited to generate a mild temperature (42 ℃) and induce a phase transition of TSL to positive release the geldanamycin and BDPII, activating the chemo-photothermal dual synergistic therapy. Then, the heat sensitivity of cancer cells was enhanced by the inhibition of HSP90, which enables considerable tumor cell death at a mild temperature produced by BDPII.)
Provide the cell viability results of BDPII-gel@TSL against normal fibroblast or normal cells.
Author reply: Thanks. we understand that the cell viability of BDPII-gel@TSL against normal cells may better reveal the selective anticancer effect. However, in the present study, we mainly focus on the efficiency in mild of BDPII-gel@TSL, and we think that the cytotoxicity assay in HeLa mey not optimal, but should be sufficient draw a conclusion that the ablation of tumor cells was improved by virtue of the chemo-photothermal dual therapeutic strategy of BDPII-gel@TSL. Also, we will utilize the appropriate and adequate experiments to explore the selective anticancer efficiency of BDPII-gel@TSL NPs in our subsequent work.
Figure 4d, calculate fluorescence intensities and analyze for statistical difference.
Author reply: Thanks. As known that the excitation and emission wavelength of Calcein-AM and PI were totally different owing to the different chemical structure, which would produce different fluorescence intensity by itself in this experiment. Based on that, the number of HeLa cells with red and green emission in Figure 4d were directly compared to evaluate the cytotoxicity for each group.